# Integrative Physiological and Molecular Insights into Drought–Induced Accumulation of Bioactive Compounds in *Clinacanthus nutans* (Burm.f.) Lindau Leaves

**DOI:** 10.3390/plants15010100

**Published:** 2025-12-29

**Authors:** Phanuwit Khamwong, Jarunee Jungkang, Usawadee Chanasut

**Affiliations:** Department of Biology, Faculty of Science, Chiang Mai University, Mueang Chiang Mai, Chiang Mai 50200, Thailand; sir.panuwit@gmail.com (P.K.); jarunee.j@cmu.ac.th (J.J.)

**Keywords:** *Clinacanthus nutans* (Burm.f.), drought stress, schaftoside, lupeol, molecular docking

## Abstract

*Clinacanthus nutans* (Burm.f.) Lindau is a medicinal plant known for its antioxidant, anti–inflammatory, and antiviral properties. Drought is a major abiotic stress affecting plant physiology and secondary metabolite biosynthesis. This study investigated the physiological and biochemical responses of *C. nutans* under drought stress. Relative water content declined with prolonged drought, while hydrogen peroxide and proline levels increased, indicating oxidative and osmotic stress. Antioxidant activities (DPPH and ABTS) peaked at days 4–8 and showed positive correlations with phenolic and flavonoid contents and triterpenoids, particularly schaftoside and lupeol. Molecular docking supported the bioactivity of drought–induced metabolites, with schaftoside and lupeol showing favorable interactions with inflammation–related targets. Multivariate analysis revealed that short–term drought enhanced stress tolerance and secondary metabolite accumulation, whereas prolonged stress reduced biosynthetic capacity and survival. These findings suggest that controlled drought exposure can enhance bioactive compound levels in *C. nutans*, supporting its potential for drought–adaptive cultivation for medicinal use.

## 1. Introduction

*Clinacanthus nutans* (Burm.f.) Lindau, commonly known as “Phaya Yor” or “snake grass,” is a Southeast Asian medicinal plant traditionally used to treat skin inflammation, herpes simplex virus infection, varicella zoster virus (shingles), and oxidative stress–related disorders [1,2]. These traditional uses are supported by pharmacological evidence demonstrating antiviral, anti–inflammatory, and antioxidant activities [3,4,5]. Phytochemical analyses revealed that *C. nutans* produces diverse classes of bioactive compounds, including phenolics, flavonoids, triterpenoids, and phytosterols [6,7]. Among them, C–glycosyl flavones such as schaftoside, orientin, vitexin, and isovitexin, together with phenolic acids like p–coumaric acid, have been reported to possess strong antioxidant and cytoprotective activities [8,9]. Moreover, triterpenoids such as lupeol contribute to immunomodulatory and anti–inflammatory effects [10,11].

Abiotic stress, particularly drought, is one of the most critical environmental factors affecting plant growth and secondary metabolism. Water deficiency induces stomatal closure, reduced photosynthetic rate, and excessive accumulation of reactive oxygen species (ROS), leading to oxidative stress and cellular damage [12,13]. To cope with this imbalance, plants activate enzymatic antioxidants—such as superoxide dismutase (SOD), catalase (CAT), and peroxidases—and accumulate non–enzymatic antioxidants like phenolics and flavonoids [14,15]. These secondary metabolites play essential roles in mitigating oxidative stress, maintaining redox homeostasis, and enhancing drought tolerance [9,16].

Several studies have indicated that moderate drought stress can stimulate the biosynthesis of secondary metabolites through the shikimate and phenylpropanoid pathways [8,17]. Such metabolite enrichment not only enhances plant stress adaptation but also increases the medicinal value of plants by improving the concentration of pharmacologically active compounds. This dual benefit highlights drought stress as a potential elicitor to improve the quality of medicinal herbs, including *C. nutans*.

Despite its recognized pharmacological significance, the molecular basis underlying the bioactivities of *C. nutans* secondary metabolites remains poorly understood. In particular, little is known about how the major phytochemicals—such as schaftoside, lupeol and p–coumaric acid—interact with target proteins involved in inflammatory and oxidative pathways. To fill this gap, computational approaches such as molecular docking have become valuable tools for predicting the interactions between phytochemicals and biomolecular targets [18,19]. This method allows in silico screening of compounds based on binding affinity, hydrogen bonding, and electrostatic complementarity, providing mechanistic insights into their potential biological roles.

Previous in silico investigations have demonstrated that plant–derived flavonoids and triterpenoids possess strong affinities toward inflammation–related enzymes, particularly cyclooxygenase–2 (COX–2) and inducible nitric oxide synthase (iNOS), which are key mediators of prostaglandin and nitric oxide production during inflammatory responses [20,21]. Consistently, ref. [22] reported that phytoconstituents of *C. nutans*, including palmitic acid and linolenyl alcohol, exhibited stable binding interactions with apoptosis– and inflammation–related proteins such as p53–binding protein Mdm2, caspase–3, and tumor necrosis factor–α (TNF–α), which correspond to the plant’s previously observed cytoprotective and antiproliferative effects. These findings collectively reinforce the pharmacological relevance of *C. nutans* metabolites and highlight their potential to modulate multiple molecular targets involved in oxidative and inflammatory signaling.

Moreover, recent advances in network pharmacology and molecular docking have revealed that flavonoids and triterpenoids commonly interact with interconnected signaling networks such as MAPK/ERK, COX–2/iNOS, and NF–κB pathways—core regulators of inflammation, oxidative stress, and apoptosis [19,23]. For example, ref. [24] demonstrated that bioactive compounds from *Toxicodendron succedaneum* could dock effectively at the ATP–binding site of extracellular signal–regulated kinase 2 (ERK2), thereby elucidating possible inhibitory mechanisms of the MAPK pathway associated with oxidative and proliferative stress. These integrative computational approaches exemplify how in silico modeling can uncover the mechanistic roles of natural compounds at the molecular level, complementing biochemical and physiological observations in plants.

Previous in silico investigations have shown that plant–derived flavonoids and triterpenoids can effectively bind to inflammation–related enzymes such as COX–2 and iNOS, key mediators of prostaglandin and nitric oxide signaling [4,21]. Similarly, in silico analysis of *C. nutans* extracts revealed that compounds like palmitic acid and linolenyl alcohol exhibit notable binding to apoptosis– and inflammation–associated targets, including Mdm2, caspase–3, and TNF–α, supporting their antioxidant and cytoprotective roles [25].

Recent advances in network pharmacology also highlight that natural flavonoids and triterpenoids act on multiple signaling pathways such as MAPK/ERK, COX–2/iNOS, and NF–κB, which regulate oxidative stress, inflammation, and apoptosis [25,26]. Integrating molecular docking with biochemical evidence from drought–induced metabolite studies, therefore, provides a multi–level perspective—from stress physiology to molecular interaction—on how *C. nutans* metabolites contribute to antioxidant and anti–inflammatory responses. This approach enhances mechanistic understanding and supports sustainable cultivation to improve the plant’s medicinal quality under limited water conditions.

Accordingly, integrating molecular docking analyses with biochemical evidence from drought–induced metabolite studies offers a multi–scale framework—from stress physiology to molecular interactions—to elucidate how *C. nutans* secondary metabolites contribute to antioxidant and anti–inflammatory activities. Therefore, the present study aims to investigate the relationship between drought stress and the accumulation of bioactive metabolites in *C. nutans*, and to predict the molecular interactions of these compounds with COX–2 and iNOS through molecular docking. This integrative strategy is expected to deepen the understanding of the biochemical and molecular basis of *C. nutans* bioactivities and to support sustainable cultivation practices that enhance its medicinal quality under water–limited conditions.

## 2. Results

### 2.1. Leaf Responses of C. nutans Under Drought Stress: Relative Water Content, H_2_O_2_, and Proline Accumulation

To confirm that the plants were subjected to water–deficit stress, this experiment evaluated both physiological and biochemical markers in *C. nutans* leaves. The physiological indicator assessed was Relative Water Content (RWC), while the biochemical markers included hydrogen peroxide (H_2_O_2_) and proline levels. Measurements were conducted on leaf samples collected from plants exposed to control conditions and drought stress for 4, 8, and 12 days.

#### 2.1.1. RWC

The RWC of *C. nutans* leaves was measured under both well–watered (control, C) and drought–stressed (DS) conditions over a period of 4, 8, and 12 days (Figure 1A). In the control group, RWC values increased gradually with time, reaching 64.83 ± 2.90%, 70.60 ± 2.05%, and 87.92 ± 9.48% on days 4, 8, and 12, respectively. In contrast, RWC in the drought–stressed group showed a consistent decline, decreasing from 48.12 ± 5.63% on day 4 to 42.64 ± 3.12% on day 8, and further dropping to 23.22 ± 3.38% by day 12.

#### 2.1.2. H_2_O_2_ and Proline Content

A marked increase in H_2_O_2_ content was observed in *C. nutans* leaves under drought stress (Figure 1B). After 4 days without watering, H_2_O_2_ levels rose sharply to 10.03 μmol/g FW and continued to increase significantly with prolonged stress, reaching 32.22 μmol/g FW by day 12. In contrast, the well–watered control group maintained relatively stable H_2_O_2_ levels, showing only a slight decline over time (8.83 to 6.68 μmol/g FW by day 4 to day 12).

Similarly, drought–stressed plants exhibited a progressive accumulation of proline in the leaves (Figure 1C), rising from 11.47 μmol/g FW on day 4 to as high as 25.99 μmol/g FW by day 12. MeaDShile, the control plants maintained consistently low proline levels (around 1 μmol/g FW) throughout the experiment. This contrasting trend highlights the role of proline as a protective osmolyte under drought stress conditions.

### 2.2. Total Bioactive Compounds in C. nutans Leaves Under Drought Stress

#### 2.2.1. Total Phenolic and Flavonoid Content

The total phenolic content (TPC) in *C. nutans* leaves reached the highest level on day 4 in both the normal condition and drought–stressed groups, with values of 5.76 and 6.12 mg/g DW, respectively (Figure 2A). After this peak, TPC gradually declined over the following days in both treatments, indicating that prolonged drought stress, as well as plant age, reduced the accumulation of phenolic compounds.

A similar pattern was observed for the total flavonoid content (TFC) (Figure 2B). The highest levels were detected on day 4, reaching approximately 0.04 mg/g DW in both the normal condition and drought–stressed plants. Thereafter, flavonoid concentration gradually declined on days 8 and 12, indicating that flavonoid biosynthesis was most active during the early phase of stress and decreased with prolonged drought exposure.

#### 2.2.2. p–Coumaric Acid, Schaftoside and Lupeol Content

p–Coumaric acid, the highest content was observed under drought stress at day 4 (DS4), with 10.55 µg/g DW, slightly higher than the normal condition group on the same day (N4: 9.37 µg/g DW), but it declined sharply thereafter, reaching only 3.49 µg/g DW by day 12 under drought stress (DS12).

For Schaftoside content reached its maximum in the drought–stressed group at day 8 (DS8), with a concentration of 28.48 µg/g DW, higher than both the normal condition and other drought treatments.

In contrast, lupeol showed a clear accumulation pattern under prolonged drought, with the maximum recorded in DS12 (13.71 µg/g DW), which was almost two–fold higher compared to the normal condition (N12: 6.38 µg/g DW).

### 2.3. Antioxidant Activity of C. nutans Leaf Extracts Under Drought Stress

The antioxidant capacity of crude extracts from *C. nutans* leaves grown under well–watered (normal condition) and drought–stressed conditions were evaluated using ABTS and DPPH assays. Antioxidant activities were expressed as Trolox equivalents (mg/g DW), and samples were analyzed at three time points: days 4, 8, and 12 (Figure 3).

In the ABTS assay, the highest antioxidant activity was observed in the drought–stressed group on day 12 (DS12), with a value of 0.097 mg/g DW, which was significantly higher than that of the normal condition group (N12: 0.076 mg/g DW, *p* < 0.01). Antioxidant capacity increased markedly during the early stage of drought (DS4 = 0.094 mg/g DW) and remained higher than in the well–watered controls throughout the experiment (C4 = 0.081, C8 = 0.076, C12 = 0.076 mg/g DW). Although slight variations were observed across sampling days, the overall trend indicated that drought–stressed plants exhibited consistently greater ABTS radical–scavenging activity than the normal condition group.

In the DPPH assay, the highest antioxidant activity was recorded in the drought–stressed group on day 4 (DS4), with a value of 0.103 mg/g DW, which was significantly higher than that of the normal condition group (N4: 0.096 mg/g DW, *p* < 0.01). Antioxidant capacity decreased steadily over time in both treatments. The values for C8, DS8, C12, and DS12 were 0.065, 0.090, 0.065, and 0.079 mg/g DW, respectively. Notably, antioxidant capacity in all drought–stressed groups (DS) remained higher than that of their respective controls, indicating that water deficit enhanced the radical–scavenging ability of *C. nutans* leaves.

### 2.4. Molecular Docking Analysis

Previous studies [21] have demonstrated that *C. nutans* extracts possess significant anti–inflammatory activity, as evidenced by the suppression of COX–2 and iNOS protein expression in Western blot analyses. Both the crude extracts and the purified compound schaftoside were reported to effectively inhibit inflammation–related signaling proteins, confirming their pharmacological relevance in inflammatory modulation. Therefore, in this study, molecular docking analysis was performed to further predict and compare the binding affinities and interaction mechanisms of schaftoside with other bioactive constituents identified in *C. nutans*, namely lupeol and p–coumaric acid [27,28], against the key inflammatory targets COX–2 and iNOS.

#### 2.4.1. Overall Binding Poses of Key Compounds

The molecular docking analysis was performed to predict the interactions of selected bioactive compounds in *C. nutans*—including schaftoside, lupeol and p–coumaric acid—with the inflammatory–related COX–2, PDB: 6COX and iNOS, PDB: 3E7G.

As shown in Figure 4, schaftoside occupied the catalytic pocket of both COX–2 and iNOS, exhibiting favorable electrostatic and hydrophobic complementarity with the residues lining the active site. The docking visualization revealed that schaftoside was well–fitted in the substrate channel of both enzymes, suggesting a conserved binding mode across the two targets.

To provide an overview of the binding strengths of the selected bioactive compounds toward the target enzymes, molecular docking analysis was performed against COX–2 and iNOS. The calculated binding affinities (kcal/mol) were used to compare the relative interaction potentials of lupeol, schaftoside, and p-coumaric acid with each target protein. The docking scores and corresponding PubChem CIDs are summarized in Table 1.

#### 2.4.2. Docking Interaction with COX–2 (PDB: 6COX)

Schaftoside was positioned deeply within the catalytic pocket, forming multiple hydrogen bonds with key residues such as Arg120, Tyr355, and Ser530 (Figure 5), which are essential for ligand anchoring and inhibition of prostaglandin synthesis. These interactions reflect strong polar complementarity and suggest that schaftoside could act as a competitive inhibitor of COX–2.

In contrast, lupeol was tightly embedded within the hydrophobic channel surrounded by Tyr385, Trp387, Val349, Leu352, and Ala527 (Figure 5), exhibiting extensive alkyl and π–alkyl interactions that stabilized the triterpenoid skeleton. The absence of hydrogen bonds indicates that its binding was predominantly driven by hydrophobic forces, similar to classical nonpolar COX–2 inhibitors.

Meanwhile, p–coumaric acid occupied a more peripheral position in the binding site, forming one to two weak hydrogen bonds with residues near the entrance channel, suggesting a relatively lower binding stability compared to schaftoside and lupeol.

#### 2.4.3. Docking Interaction with iNOS (PDB: 3E7G)

Schaftoside was located within the catalytic tunnel, forming multiple conventional and water–mediated hydrogen bonds with key residues such as Arg381, Met120, Ala282, and Trp89 (Figure 6), as well as interactions with solvent molecules (HOH4023–4061). These extensive hydrogen–bonding networks stabilize the ligand in the binding channel and suggest a strong inhibitory potential toward nitric oxide production.

Lupeol was positioned in a hydrophobic cavity surrounded by Arg381, Asp385, and Tyr491, forming π–sigma and weak hydrogen–bond interactions with surrounding residues and water molecules (HOH4024–4146) (Figure 6). The binding orientation of lupeol indicates a predominantly hydrophobic stabilization pattern, similar to its interaction mode observed with COX–2.

In contrast, p–coumaric acid interacted through a limited number of hydrogen bonds with Glu494, Trp496, and Asn261, and π–π or π–alkyl stacking with Phe286. These contacts occurred near the entrance region of the substrate channel, suggesting a relatively lower affinity compared with schaftoside and lupeol.

## 3. Discussion

The present study demonstrated that drought stress significantly influenced the accumulation of secondary metabolites and antioxidant activities in *C. nutans* leaves. Both total phenolic content (TPC) and total flavonoid content (TFC) increased markedly at the early stage of water deprivation (day 4), particularly in the DS4 group, before declining under prolonged stress. Similar responses have been reported in several medicinal and aromatic plants, where moderate drought stress enhances phenolic and flavonoid accumulation through stress–induced activation of secondary metabolic pathways, contributing to improved antioxidant capacity and cellular protection against oxidative damage [29,30].

Such transient increases in secondary metabolites under mild water deficit are commonly attributed to the role of reactive oxygen species as signaling molecules that upregulate phenylpropanoid metabolism. However, prolonged drought stress may exceed the plant’s adaptive threshold, leading to metabolic inhibition and reduced biosynthesis of phenolic compounds, which explains the decline in TPC and TFC observed at later stages of stress.

Schaftoside, the major C–glycosyl flavone in *C. nutans*, showed pronounced accumulation in DS4, consistent with the findings of Limpanich [21], who reported its role in scavenging ROS and maintaining cellular homeostasis during stress. In parallel, the triterpenoid lupeol also increased substantially under drought conditions, suggesting its involvement in protective responses. Previous studies have shown that lupeol mitigates oxidative and inflammatory damage through modulation of redox–sensitive signaling pathways and stabilization of cellular membranes [28,31]. The co–accumulation of schaftoside and lupeol therefore reflects a coordinated antioxidant defense strategy involving both phenolic and triterpenoid metabolites.

The enhanced antioxidant activities observed in DS4 extracts further support this relationship, indicating that increased levels of schaftoside and lupeol were directly associated with improved radical–scavenging capacity, as similarly reported in other medicinal plants [32,33]. The simultaneous rise in proline and H_2_O_2_ also suggests that *C. nutans* experienced both osmotic and oxidative stress, prompting biochemical adjustments comparable to those documented in *Duranta erecta* [34].

Molecular docking analysis provided additional insights into the mechanistic relevance of these drought–induced metabolites. Schaftoside and lupeol exhibited the strongest binding affinities (−9.2 kcal/mol) toward COX–2 and iNOS, forming stable interactions with essential catalytic residues. Schaftoside established multiple hydrogen bonds—Arg120, Tyr355, and Ser530 in COX–2 and Arg381, Met120, and Ala282 in iNOS—suggesting potential inhibition of prostaglandin and nitric oxide synthesis. In contrast, lupeol interacted primarily through hydrophobic and π–alkyl contacts with residues such as Tyr385 and Leu352 in COX–2 and Tyr491 in iNOS, stabilizing its triterpenoid backbone within the enzyme cavities.

Although p–coumaric acid displayed lower affinity (−6.2 kcal/mol), it formed hydrogen–bond and π–stack interactions with Glu494 and Trp496 near the entrance region of iNOS, suggesting a minor supportive role in binding stabilization. These results indicate that *C. nutans* flavonoids and triterpenoids may act through complementary inhibitory mechanisms: polar compounds like schaftoside anchor at catalytic residues, whereas non–polar lupeol reinforces hydrophobic occupancy within the active sites. Similar flavonoid–enzyme interaction patterns have been described previously [18], and structural features such as the C2–C3 double bond and hydroxyl substitutions at C3′, C4′, and C5 are known to enhance antioxidant and anti–inflammatory activities [35].

The clear separation between the normal and drought–stressed groups observed in the redundancy analysis (RDA) reflects coordinated physiological and biochemical responses of *Clinacanthus nutans* under water deficit. Drought stress induces oxidative imbalance, as evidenced by increased H_2_O_2_ levels, together with the accumulation of osmolytes such as proline and stress–responsive secondary metabolites, which collectively contribute to osmotic adjustment and redox homeostasis [36].

Plant responses to drought operate as interconnected processes affecting stomatal regulation, photosynthesis, and carbon metabolism, ultimately activating antioxidant defense systems [37]. In addition, drought stress commonly enhances the biosynthesis of phenolic compounds and flavonoids, which function in reactive oxygen species scavenging and cellular protection. The co–variation in these parameters revealed by RDA supports the view that drought stress triggers an integrated stress–adaptation strategy rather than isolated physiological responses [38].

Physiological responses supported these multivariate patterns (Figure 7): RWC declined steadily under drought, while H_2_O_2_ and proline increased, confirming oxidative and osmotic stress conditions. Correspondingly, TPC, TFC, and antioxidant activities (DPPH and ABTS) peaked at day 4, emphasizing the role of short–term drought in activating antioxidant defenses.

At the level of individual metabolites, schaftoside accumulated predominantly during early drought (DS4–DS8), consistent with its ROS–scavenging properties. p–Coumaric acid reached its maximum at day 4 but declined sharply under extended drought. In contrast, lupeol exhibited continuous and substantial accumulation throughout the drought period, peaking at DS12—nearly twice that of the normal condition—indicating its importance in late–stage drought adaptation. Docking results further supported this functional role: lupeol demonstrated strong and stable interactions with COX–2 and iNOS, comparable to schaftoside, suggesting a potential inhibitory effect on prostaglandin and nitric oxide synthesis. These predicted activities are consistent with the well–established anti–inflammatory and antioxidant properties of lupeol.

## 4. Materials and Methods

### 4.1. Plant Material and Experimental Design

Healthy *Clinacanthus nutans* (Burm.f.) Lindau cuttings (~15 cm in length) were collected from mature mother plants and propagated in plastic pots (15 cm diameter). Plants were grown in a uniform potting substrate composed of soil, compost, and coconut husk at a ratio of 2:1:1 (*v*/*v*/*v*). All plants were cultivated under controlled greenhouse conditions at Chiang Mai University, Thailand (18°47′ N, 98°57′ E), with a photoperiod of 12 h light/12 h dark, light intensity of 3000 ± 200 lux, average temperature of 30 ± 2 °C, and relative humidity of 75 ± 5%. Well–watered control (C): plants watered daily to maintain 80–90% field capacity.

After 8 weeks of growth, uniform and healthy plants were randomly assigned to experimental groups using a completely randomized design. The experiment consisted of two main treatments:(1)Normal condition (N): plants were irrigated daily with a fixed volume of 5 mL water per pot to maintain consistently well–watered conditions.(2)Drought stress (DS): irrigation was completely withheld for 4, 8, or 12 consecutive days to induce progressive water deficit.

To allow direct comparison between drought–stressed and control plants at the same developmental stage, control groups were sampled at corresponding time points (4, 8, and 12 days). Each treatment consisted of three biological replicates (*n* = 3), with each replicate comprising five individual plants. Leaf samples were collected at the end of each drought period (day 4, day 8, and day 12) for physiological and biochemical analyses.

#### Extraction of Plant Materials

For dried sample extraction, finely powdered dried leaves of *C. nutans* were extracted with 80% (*v*/*v*) ethanol at a sample–to–solvent ratio of 1:50 (*w*/*v*). The mixture was sonicated for 60 min and then filtered through Whatman No. 1 filter paper. The extraction process was repeated twice on the residue obtained from the previous filtration. The combined filtrates were concentrated under reduced pressure using a rotary evaporator (Rotavapor R–210, Buchi, Flawil, Switzerland) and subsequently lyophilized (Virtis Benchtop K Freeze Dryer, SP Industries, Warminster, PA, USA). The dried extracts were stored at −20 °C until further analysis. (Sarega et al. [27]).

For fresh sample extraction, freshly collected leaves were macerated in 80% (*v*/*v*) ethanol at a ratio of 1:10 (*w*/*v*) and subjected to reflux extraction using a magnetic stirrer and temperature–controlled water bath. All experiments were performed in triplicate. After extraction, the samples were filtered, concentrated using a rotary evaporator (Rotavapor R–210, Buchi, Switzerland) at approximately 60 °C, weighed, and stored at −20 °C until further use. (Cheng et al. [28]).

### 4.2. Assessment of Drought Stress Tolerance

#### RWC Method

Leaf RWC was determined according to [39,40] by measuring fresh weight (FW), turgid weight (TW), and dried weight (DW), using the formula:$$RWC\;(\%)=\frac{FW-DW}{TW-DW}\times 100$$

### 4.3. Physiological and Biochemical Measurements

#### 4.3.1. H_2_O_2_ Content

H_2_O_2_ levels were measured following the method of Nguyen et al. [31] with modifications. Leaf tissue (0.3 g) was homogenized in 0.1% trichloroacetic acid (TCA) and centrifuged; the supernatant was reacted with potassium iodide and absorbance was read at 390 nm.

#### 4.3.2. Proline Content

Proline accumulation was quantified using the method of Haida et al. [32]. Leaf tissue was homogenized in sulfosalicylic acid, reacted with acid ninhydrin, and the absorbance was read at 520 nm.

### 4.4. Colorimetric Determination

#### 4.4.1. Total Phenolic Contents

TPC was determined using the Folin–Ciocalteu method with minor modifications [41]. Briefly, 0.5 mL of leaf extract was mixed with 2.5 mL of 10–fold diluted Folin–Ciocalteu’s reagent (*v*/*v*) and allowed to stand at room temperature in the dark for 5 min. Subsequently, 2.0 mL of 7.5% (*w*/*v*) sodium carbonate (Na_2_CO_3_) solution was added, mixed thoroughly, and incubated in the dark at 25–30 °C for 90 min. The absorbance was measured at 760 nm using a UV–Vis spectrophotometer (blank: deionized water). Gallic acid was used to construct a standard calibration curve, and results were expressed as mg gallic acid equivalent per g DW (mg GAE/g DW).

#### 4.4.2. Total Flavonoid Contents

TFC was analyzed using the aluminum chloride colorimetric assay [22]. A 0.5 mL aliquot of extract was mixed with 0.3 mL of 5% sodium nitrite (NaNO_2_), shaken gently, and incubated for 5 min at room temperature. Then, 0.3 mL of 10% aluminum chloride (AlCl_3_) solution was added, mixed, and incubated for 6 min, followed by the addition of 2.0 mL of 4% sodium hydroxide (NaOH) solution. The mixture was shaken thoroughly and kept in the dark for 15 min. Absorbance was recorded at 510 nm (blank: deionized water). Quercetin was used as the calibration standard, and results were expressed as mg quercetin equivalent per g DW (mg QE/g DW).

### 4.5. HPLC/DAD Analysis [21,42]

The quantification of schaftoside and p–coumaric acid in *C. nutans* leaf extracts was performed using a high–performance liquid chromatography system (HPLC) (Agilent 1260 Infinity II, Agilent Technologies, Waldbronn, Germany) equipped with a diode array detector (DAD) and an autosampler. Separation was carried out on a Zorbax Eclipse XDB–C18 column (4.6 × 150 mm, 5 μm; Agilent Technologies) under specific chromatographic conditions for each compound.

For schaftoside, the detection wavelength was set at 340 nm. The column temperature was maintained at 35 °C, with a mobile phase consisting of solvent A (acetonitrile) and solvent B (0.1% formic acid in deionized water). The gradient elution started at 12% A (0–15 min), increased to 13–15% A (15–25 min), at a flow rate of 0.8 mL/min. The injection volume was 10 μL.

For p–coumaric acid, detection was performed at 310 nm with the column temperature at 30 °C. The mobile phase consisted of solvent A (0.1% orthophosphoric acid in deionized water), solvent B (0.1% orthophosphoric acid in methanol), and solvent C (0.1% orthophosphoric acid in acetonitrile) in the ratio of 65:25:10 (*v*/*v*/*v*). The isocratic flow rate was maintained at 1.0 mL/min, with an injection volume of 10 μL.

### 4.6. Gas Chromatography (GC–FID) Analysis [42]

The analysis of lupeol was performed using a gas chromatograph (Agilent 7890B, Agilent Technologies, Santa Clara, CA, USA) equipped with a flame ionization detector (FID). The injection was carried out in split mode at a ratio of 10:1, with an injection volume of 1 µL. Helium was used as the carrier gas at a constant flow rate of 1 mL min^−1^. The injector temperature was set at 250 °C, and the detector temperature was maintained at 270 °C. The oven temperature program was initiated at 100 °C, then increased at a rate of 4 °C min^−1^ to 270 °C, and held for 30 min, resulting in a total run time of 125 min. Separation was achieved on an HP–5 capillary column (30 m × 0.32 mm i.d., film thickness 0.25 µm). The temperature of the auxiliary heater was maintained at 280 °C. Under these conditions, lupeol was detected at a retention time (RT) of approximately 97 min.

### 4.7. Molecular Docking Analysis

Molecular docking was performed following the method of Ismail et al. [25] with slight modifications to predict the binding interactions between *C. nutans* metabolites and inflammation–related targets, COX–2; PDB ID: 6COX and iNOS; PDB ID: 3E7G. Protein structures were retrieved from the Protein Data Bank, and ligands were obtained from PubChem in SDF format. Ligand conversion and energy minimization were carried out using Open Babel GUI, while docking simulations were performed in PyRx integrated with AutoDock Vina. The grid box was set to cover the native ligand binding site, and binding affinities were recorded in kcal/mol. Visualization and interaction analyses were conducted using Discovery Studio Visualizer, version 2025, Dassault Systèmes BIOVIA focusing on hydrogen bonding and hydrophobic interactions with active site residues.

## 5. Conclusions

This study demonstrated that drought stress markedly influenced the physiological and biochemical responses of *C. nutans*, resulting in distinct changes in secondary metabolite accumulation. Moderate to prolonged drought elicited coordinated biochemical adjustments that enhanced the levels of key pharmacologically active compounds—particularly lupeol and schaftoside—thereby reinforcing both stress tolerance and medicinal potential.

By integrating physiological measurements, metabolite profiling, and molecular docking evidence, this work provides deeper insight into the biochemical and molecular foundations underlying the bioactivities of *C. nutans*. These findings highlight controlled drought exposure as a practical approach for improving metabolite production and support sustainable cultivation strategies aimed at enhancing the plant’s medicinal quality under water–limited conditions.

## Figures and Tables

**Figure 1 plants-15-00100-f001:**
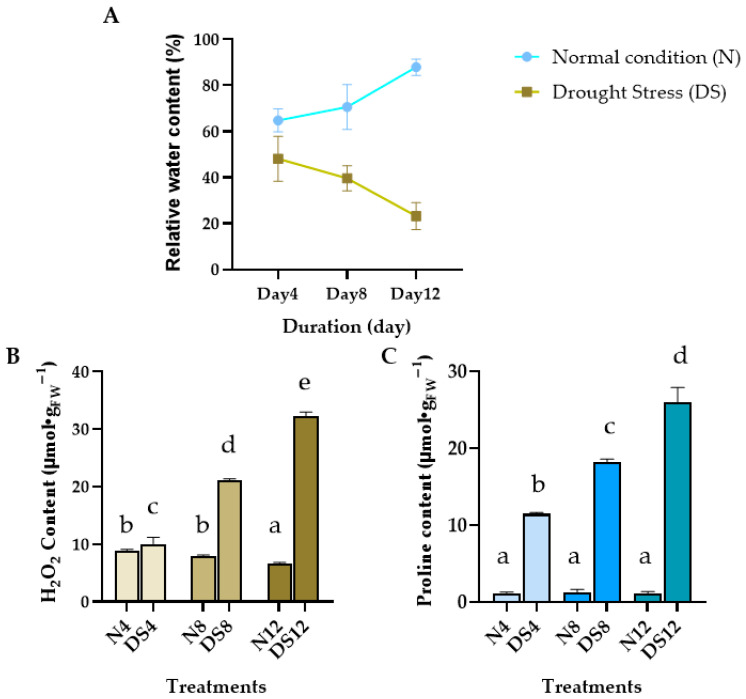
Physiological and biochemical responses of *Clinacanthus nutans* leaves under drought stress. (**A**) Relative water content (RWC) under normal conditions (N) and drought–stressed (DS) conditions at 4, 8, and 12 days; (**B**) Hydrogen peroxide (H_2_O_2_) content; (**C**) Proline accumulation in leaves during drought exposure. Data are presented as mean ± standard error (*n* = 6). Different letters indicate significant differences at *p* < 0.01 according to Duncan’s multiple range test (DMRT).

**Figure 2 plants-15-00100-f002:**
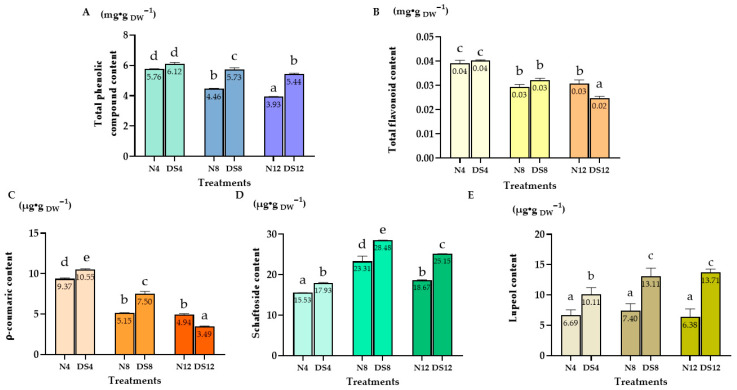
Accumulation of secondary metabolites in *Clinacanthus nutans* leaves under drought stress. (**A**) Total phenolic content (TPC); (**B**) Total flavonoid content (TFC); (**C**) p–Coumaric acid content; (**D**) Schaftoside content; (**E**) Lupeol content under normal conditions (N) and drought–stressed (DS) conditions at 4, 8, and 12 days. Data are presented as mean ± standard error (*n* = 3 for individual compounds; *n* = 6 for TPC and TFC). Different letters indicate significant differences at *p* < 0.01 according to Duncan’s multiple range test (DMRT).

**Figure 3 plants-15-00100-f003:**
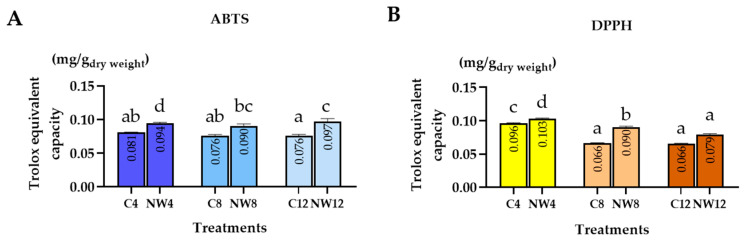
Antioxidant activity of crude extracts from *C. nutans* leaves grown under normal conditions (N) and drought–stressed (DS) conditions for 4, 8, and 12 days, as evaluated by (**A**) ABTS and (**B**) DPPH. Data were expressed as mean ± standard error (*n* = 6 ± SE). Different letters indicate statistically significant differences at *p* < 0.01, as determined by DMRT.

**Figure 4 plants-15-00100-f004:**
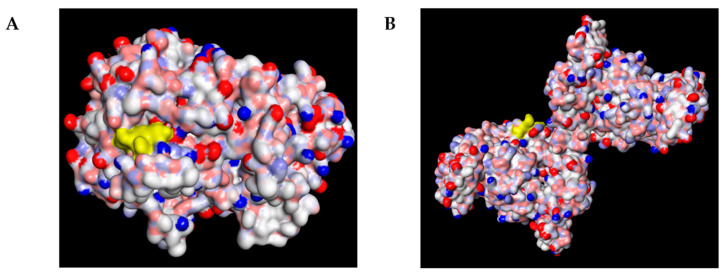
Molecular docking and interaction pattern analysis of schaftoside with COX–2 and iNOS: (**A**) Overall binding pose of schaftoside (yellow) within the catalytic pocket of COX–2 (PDB ID: 6COX), shown as an electrostatic surface (red = negative, blue = positive, white = neutral), indicating favorable electrostatic and hydrophobic complementarity. (**B**) Overall binding pose of schaftoside (yellow) within the active–site channel of iNOS (PDB ID: 3E7G), displayed as an electrostatic surface, suggesting a conserved binding mode across both targets.

**Figure 5 plants-15-00100-f005:**
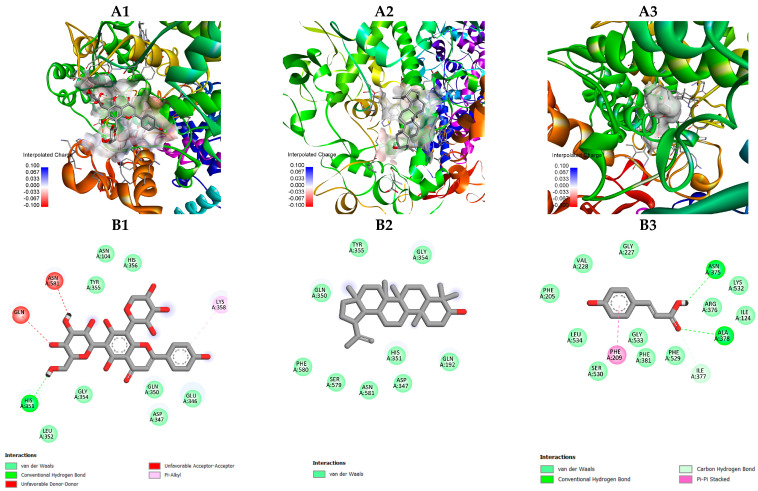
Molecular docking models of selected compounds from *C. nutans* with COX–2 (PDB ID: 6COX): (**A1**–**A3**) Three–dimensional binding poses of schaftoside (**A1**), lupeol (**A2**), and p–coumaric acid (**A3**) within the COX–2 catalytic pocket. (**B1**–**B3**) Two–dimensional interaction diagrams illustrating interactions with key active–site residues. Schaftoside forms hydrogen bonds with Arg120, Tyr355, and Ser530; lupeol exhibits predominantly hydrophobic interactions with Tyr385 and Leu352, while p–coumaric acid shows weaker interactions near the entrance of the active site.

**Figure 6 plants-15-00100-f006:**
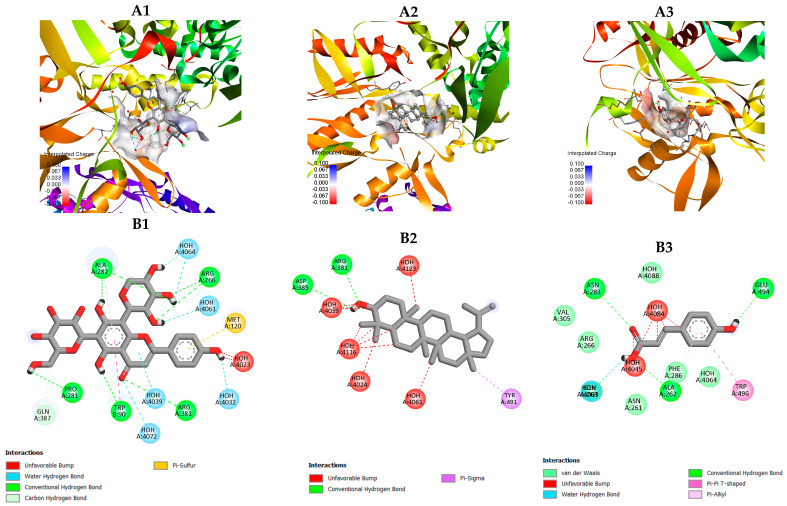
Molecular docking models of selected compounds from *C. nutans* with iNOS (PDB ID: 3E7G): (**A1**–**A3**) Three–dimensional binding poses of schaftoside (**A1**), lupeol (**A2**), and p–coumaric acid (**A3**) within the iNOS active–site channel, showing surface charge distribution (red = negative, blue = positive). (**B1**–**B3**) Two–dimensional interaction diagrams of schaftoside (**B1**), lupeol (**B2**), and p–coumaric acid (**B3**) with surrounding amino acid residues of iNOS. Schaftoside forms multiple conventional and water–mediated hydrogen bonds with Arg381, Met120, and Ala282; lupeol exhibits hydrophobic and π–sigma interactions with Arg381 and Tyr491, while p–coumaric acid interacts through hydrogen bonds and π–alkyl stacking with Glu494, Trp496, and Phe286 near the enzyme’s entrance region.

**Figure 7 plants-15-00100-f007:**
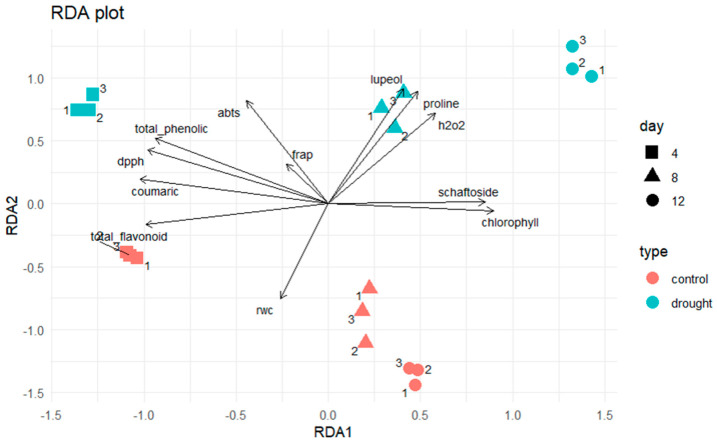
Redundancy analysis (RDA) illustrates the correlations among physiological and biochemical traits in *C. nutans* leaves under normal conditions (N) and drought stress (DS) at 4, 8, and 12 days. Replicates were color–coded according to treatment groups. The DS12 group shows strong correlations with hydrogen peroxide (H_2_O_2_) and proline, whereas DS4 and DS8 are associated with total phenolics, flavonoids, and schaftoside, indicating distinct physiological and biochemical responses to drought severity. Detailed chlorophyll and FRAP results are presented in the Appendix A.

**Table 1 plants-15-00100-t001:** Binding affinities of selected bioactive compounds from *C. nutans* with COX–2 (PDB: 6COX) and iNOS (PDB: 3E7G), including PubChem CID and docking scores (kcal/mol).

Target	Compound	PubChem CID	Affinity (kcal/mol)
COX–2 (PDB: 6COX)	Lupeol	259846	−6.7
	Schaftoside	442658	−6.5
	ρ–Coumaric acid	637542	−6.6
iNOS (PDB: 3E7G)	Lupeol	259846	−9.2
	Schaftoside	442658	−9.2
	ρ–Coumaric acid	637542	−6.2

## Data Availability

The datasets used and analyzed during the current study are available from the corresponding author on reasonable request.

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
