# Peer review of "Integrative Physiological and Molecular Insights into Drought–Induced Accumulation of Bioactive Compounds in *Clinacanthus nutans* (Burm.f.) Lindau Leaves"

_plants, 2025, doi:10.3390/plants15010100_

Round 1
Reviewer 1 Report
Comments and Suggestions for Authors
Review of the paper entitled
Integrative Physiological and Molecular Insights into Drought- Induced Accumulation of Bioactive Compounds in Clinacanthus nutans Leaves
The manuscript describes several tests, measurements of antioxidant capacities, investigated the physiological and biochemical responses of C. nutans under drought stress.
This is an interesting paper, the study is relevant, and the author contribute to this field of research.
The title reflects the main purpose of the manuscript precisely. The methods used and the results were well described, the conclusions are relevant.
However, authors need to make some important corrections
The manuscript was not prepared carefully.
In the affiliations points 1 and 2 are identical, but author no. 3 does not have the affiliation listed. Please correct it according to the Instructions for authors.
Throughout the entire paper, including in the title, the correct name of the species should be used - Clinacanthus nutans (Burm.f.) Lindau
In according to the Instructions for authors, after the introduction, the materials and methods section follows. Please correct it.
In the Results section, in the explanations for Figures 3-5, the role of letters a-e should be explained in more detail.
The authors need to check the list of references and correct it according to the Instructions for authors.
Author Response
We thank the reviewer for the valuable comments and suggestions. All issues raised have been carefully addressed, and the manuscript has been revised accordingly in compliance with the Instructions for Authors.
Reviewer 2 Report
Comments and Suggestions for Authors
This manuscript concentrated on the investigation of physiological and biochemical responses to drought stress in Clinacanthus nutans, and the authors combined with enzyme activity detection, molecular docking, HPLC/DAD, and GC-FID analyses to reveal the responding mechanism agianst abiotic stress. In general, those results are of innovation and significance for drought studies, while some minor modifications must be put forward first.
(1) Please add the important data in the Abstact, and highlight the significance of this study in Abstact.
(2) Suggest to revise C and NW to N and DS, since the suthors utulized the normal condication and drought stress to deal with the plants.
(3) Suggest to remove some figurs into supplementary files, since less than 6 figures are sultable for publciation.
(4) Suggest to revise the Table 1 into the three-wire format.
(5) Line 394, revise the reference format.
(6) Normally, two or more varieties should be chosen for physiological and biochemical responses to drought stress, while the authors just chosed one species. We might believe the number of species is not enough.
(7) Less than 40 references were cited in this manuscript, and suggest to add more references.
Author Response
Response to Comment (6):
We appreciate the reviewer’s valuable suggestion regarding the inclusion of multiple species or varieties for drought stress studies. In this case, the present work represents a focused and independent experiment derived from our previous research (DOI: 10.3390/ijms26136029). The overarching research project was specifically designed to investigate the physiological, biochemical, and molecular responses of Clinacanthus nutans (Burm.f.) Lindau, with particular emphasis on the bioactivity and mechanistic roles of its extracts and major metabolites.
Therefore, only a single plant species was intentionally selected in order to maintain experimental consistency and to align with the defined objectives and scope of the funded research project. This targeted approach allowed for a more detailed and in-depth analysis of drought-induced metabolic and physiological changes in C. nutans, rather than a comparative screening across multiple species. We have clarified this rationale in the revised manuscript to avoid potential misunderstanding.
Reviewer 3 Report
Comments and Suggestions for Authors
The manuscript entitled “Integrative Physiological and Molecular Insights into Drought-Induced Accumulation of Bioactive Compounds in Clinacanthus nutans Leaves” addresses Clinacanthus nutans, a medicinal plant known for its antioxidant, anti-inflammatory, and antiviral properties. The study investigated the physiological and biochemical responses of this species under drought stress. Reduced water availability decreased the relative water content of the leaves and increased stress markers such as hydrogen peroxide and proline. Between 4 and 8 days of drought, the authors observed higher antioxidant activity and accumulation of bioactive compounds, including phenolics, flavonoids, and triterpenoids. They concluded that moderate drought can stimulate the production of medicinal compounds, whereas prolonged drought reduces the plant’s biosynthetic capacity and survival. Overall, this is a well-written study with solid results and aligned with the journal’s scope. Below are some suggestions to further improve the manuscript:
- L85: in silico should be italicized; revise throughout the manuscript.
- The unit “μmol/g” should be corrected to “μmol g⁻¹”; adjust all occurrences.
- Some in-text citations are incorrectly formatted, such as in line 385. The correct format is: “(Name Author [27])”. Please revise all citations.
- Several figure captions present formatting issues, particularly Figures 6, 7, 8, and 11. Please review all captions to ensure consistency and proper formatting.
- The methodology lacks essential information that compromises the reproducibility of the experiment. The procedure used to determine field capacity is not described, even though the control treatment is maintained at 80–90% of this parameter. It is also not explained why drought stress is considered established after 4 days without irrigation, as there is no monitoring of substrate moisture or the plants’ water status.
- The description of drought imposition is limited to “withheld for 4, 8, or 12 days,” without specifying whether irrigation was completely suspended, whether plants were saturated beforehand, or whether environmental factors were controlled. Additionally, the substrate is insufficiently described: the “soil:compost (1:1)” mixture does not indicate the soil type, compost composition, or physical properties, which makes replication impossible.
- Clarify how the evaluations performed at days 4, 8, and 12 were conducted: were the plants destructively sampled at each time point and replaced with new samples?
- In summary, details on field capacity determination, irrigation management, and substrate characterization are missing, making the methodology unclear and lacking reproducibility.
Author Response
Response to Reviewer
We thank the reviewer for the valuable comments regarding the clarity and reproducibility of the experimental methodology. We appreciate the opportunity to further clarify the experimental design and drought imposition strategy.
Field Capacity Determination
Regarding field capacity, preliminary experiments were conducted in our laboratory prior to the main study to evaluate the water-holding capacity of the potting substrate used (soil:compost mixture at the specified ratio). These preliminary observations indicated that the substrate exhibited a very high water-holding capacity, exceeding the actual water requirement of Clinacanthus nutans under the given cultivation conditions. Consequently, maintaining plants at a defined percentage of field capacity was not considered an appropriate indicator of plant water status in this system.
Based on these findings, the experimental design was adjusted to focus on plant-based water status indicators rather than soil-based measurements. The irrigation regime for the normal condition was therefore standardized using a fixed daily water volume under identical growth conditions, allowing consistent comparison between treatments. The conceptual basis of this approach has been previously investigated and reported in a preliminary undergraduate study conducted under the same cultivation conditions. The present study adopted this plant-centered approach to assess drought stress responses.
Justification for Drought Duration and Establishment of Stress
The selection of drought durations (4, 8, and 12 days without irrigation) was based on preliminary drought trials in which longer periods of water withholding were tested followed by rewatering. These trials demonstrated that plants subjected to water deprivation beyond 12 days were largely unable to recover after re-irrigation, indicating that they had exceeded the permanent wilting point. Therefore, 12 days was defined as the maximum drought duration that allowed assessment of severe stress responses while avoiding complete plant mortality.
Drought stress was considered established at day 4 because this time point represents an early stage of water deficit at which clear physiological and biochemical responses were detectable. In the present study, plant water status was monitored using relative water content (RWC), together with hydrogen peroxide (H₂O₂) and proline accumulation, which are widely accepted indicators of oxidative and osmotic stress in plants. These parameters provided direct evidence of drought-induced stress progression across all sampling time points.
Drought Imposition and Environmental Control
For drought-stressed treatments, irrigation was completely withheld for 4, 8, or 12 consecutive days. Plants were not artificially saturated prior to drought imposition; instead, all treatments were initiated under identical normal growth conditions to ensure comparability. Environmental factors, including light intensity, photoperiod, temperature, and relative humidity, were maintained consistently throughout the experimental period.
Substrate Description
All plants were grown in the same potting substrate prepared using a fixed formulation, and identical pot size and cultivation conditions were applied across all treatments. Although detailed physical properties of the substrate were not measured, the use of a uniform substrate composition and controlled growth conditions ensured consistent treatment comparisons within the experiment.
Sampling Strategy
Leaf sampling at days 4, 8, and 12 was conducted using destructive sampling, which is standard practice for biochemical extraction and analysis. To accommodate this approach, plants were continuously propagated and maintained in the laboratory to replace sampled individuals. Each sampling time point therefore represented an independent set of plants grown under identical conditions.
Supplementary Information
To provide additional transparency and support for the experimental design, the full master’s thesis containing detailed methodological background has been included as a supplementary file. We apologize that this document is available in Thai; however, it is provided to enhance clarity and reproducibility for interested readers.
